# NBSTRN Tools to Advance Newborn Screening Research and Support Newborn Screening Stakeholders

**DOI:** 10.3390/ijns9040063

**Published:** 2023-10-30

**Authors:** Kee Chan, Zhanzhi Hu, Lynn W. Bush, Heidi Cope, Ingrid A. Holm, Stephen F. Kingsmore, Kevin Wilhelm, Curt Scharfe, Amy Brower

**Affiliations:** 1American College of Medical Genetics and Genomics, Bethesda, MD 20814, USA; abrower@acmg.net; 2Department of Systems Biology, Columbia University, New York, NY 10032, USA; zh2425@cumc.columbia.edu; 3Division Genetics and Genomics, Boston Children’s Hospital Center, Boston, MA 02115, USA; lynn.bush@childrens.harvard.edu (L.W.B.); ingrid.holm@childrens.harvard.edu (I.A.H.); 4Department of Pediatrics and Center for Bioethics, Harvard Medical School, Boston, MA 02115, USA; 5GenOmics and Translational Research Center, RTI International, Research Triangle Park, NC 27709, USA; hcope@rti.org; 6Rady Children’s Institute for Genomic Medicine, San Diego, CA 92123, USA; skingsmore@rchsd.org; 7Department of Molecular and Human Genetics, Baylor College of Medicine, Houston, TX 77030, USA; kevin.wilhelm@bcm.edu; 8Department of Genetics, Yale School of Medicine, New Haven, CT 06510, USA; curt.scharfe@yale.edu

**Keywords:** newborn screening, genome sequencing, ethical, legal, social implications (ELSI), NBSTRN, public health, databases, online tools, Recommended Uniform Screening Panel (RUSP)

## Abstract

Rapid advances in the screening, diagnosis, and treatment of genetic disorders have increased the number of conditions that can be detected through universal newborn screening (NBS). However, the addition of conditions to the Recommended Uniform Screening Panel (RUSP) and the implementation of nationwide screening has been a slow process taking several years to accomplish for individual conditions. Here, we describe web-based tools and resources developed and implemented by the newborn screening translational research network (NBSTRN) to advance newborn screening research and support NBS stakeholders worldwide. The NBSTRN’s tools include the Longitudinal Pediatric Data Resource (LPDR), the NBS Condition Resource (NBS-CR), the NBS Virtual Repository (NBS-VR), and the Ethical, Legal, and Social Issues (ELSI) Advantage. Research programs, including the Inborn Errors of Metabolism Information System (IBEM-IS), BabySeq, EarlyCheck, and Family Narratives Use Cases, have utilized NBSTRN’s tools and, in turn, contributed research data to further expand and refine these resources. Additionally, we discuss ongoing tool development to facilitate the expansion of genetic disease screening in increasingly diverse populations. In conclusion, NBSTRN’s tools and resources provide a trusted platform to enable NBS stakeholders to advance NBS research and improve clinical care for patients and their families.

## 1. Introduction

Current newborn screening (NBS) research, implementation, and evaluation is a collaborative effort among diverse stakeholders such as healthcare professionals, researchers, public health representatives, families, advocacy groups, and numerous others. The federal Health Resources and Services Administration (HRSA) maintains the Recommended Uniform Screening Panel (RUSP) for NBS [1]. Conditions are selected for nominations and inclusion on the RUSP based on evidence that supports the potential net benefit of screening, the ability of states to screen for the disorder, and the availability of effective treatments and clinical management. Adding any new condition to the RUSP is a highly selective and lengthy process.

As of January 2023, the RUSP lists 63 conditions recommended for NBS. Each state’s public health service is responsible for determining which conditions to include in their NBS program, as states are not required to screen for all conditions. Most states screen for the majority of RUSP disorders and some opt to screen for additional conditions. Currently, 81 different conditions are screened across the U.S., with a state-specific high of 71 and a low of 32 [2]. It is important to note that state programs count disorders in different ways and these numbers are likely not comparable [3].

Advances in technologies for the screening, diagnosis, and treatment of disorders in newborns can potentially increase the number of conditions that can be addressed through NBS. Even so, the expansion of NBS is currently a slow process—it takes an average of 9.5 years for the nationwide adoption of NBS for a new RUSP-listed condition [3]. To better understand the factors that most influence NBS expansion—both positively and negatively—the Newborn Screening Translational Research Network (NBSTRN) conducted an NBS Expansion Study to identify weaknesses and gaps in the current NBS system [3]. The study used a multi-layer strategy to gather input from various NBS stakeholders and resources, including a workshop of NBS experts, a survey of clinicians, a literature review, and a review of online resources and key efforts.

The NBSTRN study identified four factors that delay or complicate NBS expansion [3]. The researchers further evaluated potential solutions to address those challenges [4].

Lack of State Pilot Data Dissemination Procedure—A potential solution is for states to have a real-time tracking system where all pilot studies and outcomes are aggregated.Limited Longitudinal Data—A potential solution is to develop a long-term follow-up data platform that captures treatment and outcome data beyond the pilot stage. Such data could be used by clinicians, state NBS programs, and families to coordinate and improve patient care.Variable Onset Conditions—A potential solution is to gather information from NBS stakeholders on recognized facilitators of and barriers to NBS expansion in cases of variable onset conditions.RUSP Process Capacity Constraints—A potential solution is to create an overarching system that collects data from NBS pilots. Such a system would provide a data archive that could be used to look for supporting evidence for the simultaneous evaluation of multiple candidate conditions.

The NBS Expansion Study highlights the need for more robust, easily accessible data tools that stakeholders can use to advance NBS capabilities, timeframes, and patient outcomes. Addressing these challenges will require innovative solutions to modernize the NBS system and make it more responsive to rapid advances in technology and the expectations of families and advocacy groups. Further, most NBS conditions represent a continuum of disease severity with variability in clinical presentation and outcomes. For example, in phenylketonuria (PKU), plasma phenylalanine (Phe) levels and tolerance to ingested Phe vary widely among affected individuals, with the milder end of the spectrum typically not requiring treatment [5], while other conditions such as the rare Pompe disease have variable rates of disease progression and different ages of onset.

Outside of the U.S., most NBS programs in Europe have modernized the screening process with new technology or algorithms with new information (i.e., modifying cut-off rates based on different platforms) and expanded their panel of conditions screened based on the screening criteria in different countries [6]. However, the decision-making oversights across the globe vary in their adoption, implementation, and resources to support the short- and long-term management of care. To facilitate the advancement and sharing of NBS research, the International Society of Neonatal Screening is in the process of establishing a database, which is available for public consultation [6]. As the cost of genome sequencing has decreased drastically over the last decade and the speed of using genomic data to inform diagnosis has increased, Australia, China, and many countries in Europe are considering the application of genomic sequencing platforms to complement newborn screening [7,8,9].

This paper describes the data tools and resources developed and implemented by the NBSTRN to facilitate NBS research and support NBS stakeholders locally and internationally. The NBSTRN’s web-based tools are accessible worldwide to share knowledge and advance the development and expansion of NBS programs locally and internationally. We provide use cases for each NBSTRN tool such as the NBS Condition Resource (NBS-CR) for information about screenable disorders, the dissemination of results from pilot programs, and the identification of candidate conditions for newborn screening across the globe. For example, international countries could use the NBS-CR to identify possible candidates for newborn screening. We also discuss new tools that are currently not available and that may be required as NBS expands to include a larger number of genetic disorders that are screened in increasingly diverse populations.

## 2. Materials and Methods

### 2.1. NBSTRN

The National Institute of Child Health and Human Development (NICHD) Hunter Kelly Newborn Screening Research Program conducts and funds research to advance NBS. Part of that work was the establishment in 2008 of the NBSTRN. The NBSTRN’s vision is to facilitate and support ground-breaking research in order to:Accelerate the understanding of early-onset genetic diseases;Increase the number of screened conditions;Foster collaboration among clinicians, families, health professionals, and other stakeholders to understand and maximize health outcomes.

Through a contract with the American College of Medical Genetics and Genomics (ACMG), the NBSTRN develops, maintains, and enhances tools, resources, and expertise in support of NBS research. The NBSTRN has grown into an international network of hundreds of participants conducting cutting-edge research, population-based pilots, and ethical, legal, and social (ELSI) studies to advance newborn screening [10].

### 2.2. NBS Stakeholders

NBS stakeholders include numerous individuals across diverse organizations. The main stakeholders that the NBSTRN works with directly are researchers, healthcare professionals, families, advocacy groups, and state NBS programs. Each of these stakeholders play an important and unique role in NBS advancement and patient outcomes.

#### 2.2.1. Researchers and Healthcare Professionals

NBS researchers, physician-scientists, and diverse healthcare professionals have backgrounds in a range of disciplines, such as specialized clinical practices, medical genetics, genetic counseling, psychology, public health, and more. The common element among these diverse groups is their desire to help advance the screening, diagnosis, and treatment of rare diseases in newborns and children. The needs of NBS researchers are also diverse and include data tools and repositories, funding opportunities, the input of experts in different areas of NBS, and other support for their research. Healthcare stakeholders have expressed a need for help translating research findings into clinical practice. Many healthcare professionals are also involved in NBS research and play key roles in both pilot studies and longitudinal studies.

#### 2.2.2. Families and Advocacy Groups

Families and advocacy groups are often interested in new research findings, especially when a condition is under consideration for RUSP listing. Families may also become interested if they learn that a condition is not on the RUSP and is not under consideration. They need a reliable source for information about the latest research efforts and discoveries, help understanding specific conditions, and information about the ethical, legal, and social issues in NBS research.

#### 2.2.3. State NBS Programs

State public health programs conduct pilot research studies and need a reliable source for other pilot study data. They also need a way to share their research findings with the NBS community.

### 2.3. Tool Development

The NBSTRN has developed data tools, resources, and expertise to facilitate and support ground-breaking NBS research, clinical care, families, communities, and program development. The NBSTRN provides resources for each segment of the NBS community [11]. This paper focuses on the NBSTRN’s data tools for NBS researchers.

## 3. Results

Four key NBSTRN data tools that researchers have come to rely upon are [12]:Longitudinal Pediatric Data Resource (LPDR);NBS Condition Resource (NBS-CR);NBS Virtual Repository (NBS-VR);ELSI Advantage.

These tools are used hundreds of times each year by various NBS stakeholders. The following sections describe these tools and present a use case of how each is being used by researchers to advance NBS capabilities and outcomes.

### 3.1. Longitudinal Pediatric Data Resource

The NBSTRN created the Longitudinal Pediatric Data Resource (LPDR) as a suite of tools that can be used to capture, store, share, analyze, and visualize data and other condition-specific information [13]. Having access to a combined data repository such as the LPDR is important because most NBS conditions are rare and accumulating enough subject data to have statistical power is often a barrier to assessing effective disease diagnosis for understanding health outcomes and the benefits of early identification and treatment.

#### 3.1.1. Genomic and Phenotypic Data

Since its launch in 2013, the LPDR has collected more than 12 million data points from nearly 9000 subjects. At present, the LPDR includes 20 studies and 134 individual conditions. Researchers can explore this information stratified by demographic and other factors.

The LPDR data have been contributed by more than 30 NBS stakeholders from numerous U.S. states and territories. More than 100 researchers and State NBS programs have used the LPDR in translational, public health, and clinical research projects. The LPDR is designed to share these teams’ new findings and to foster the secondary use of the original data sets.

The LPDR is used by research teams conducting longitudinal studies of the RUSP and candidate conditions, exploring the use of genome sequencing in the newborn period, and conducting NBS pilots of candidate conditions. Case-level, de-identified data sets are also included in the LPDR and are available for secondary use by the research community.

#### 3.1.2. Common Data Elements

The LPDR also contains sets of question-and-answer choices called common data elements (CDEs) for RUSP and candidate conditions. To date, more than 24,000 CDEs for 118 conditions have been included in the LPDR. The result is a catalog of CDEs that allows researchers and public health teams to systematically collect, analyze, and share data across the NBS community. The CDEs are also used to create data dictionaries and electronic case report forms (eCRFs) organized chronologically to match a subject’s longitudinal care record over time.

Researchers can find the LPDR useful in many ways, such as:Investigating unique data sets;Collaborating with leading investigators;Designing a study using validated CDEs.

The LPDR provides researchers with access to data and information from across the lifespan of NBS-identified newborns, which they can potentially use to improve outcomes for NBS patients and families. Visitors to NBSTRN.org can select the LPDR link within the Data Tools menu and request access to the database.

#### 3.1.3. LPDR Use Cases

##### IBEMC Database Use Case

The LPDR contains the Inborn Errors of Metabolism Information System (IBEM-IS) data repository for IBEMC longitudinal natural history data [14,15]. A recent study used IBEM-IS data available in the LPDR to characterize and classify novel variants across 32 diseases [16]. The researchers aggregated data from 982 IBEM-IS participants (51.6%) who had confirmatory genotyping analysis to assess the novelty of the recorded variants. A comparison of the IBEMC-reported variants against variants reported in the ClinVar database [17] identified 161 novel variants not published in ClinVar.

Next, the researchers used IBEMC phenotypic data, also contained in the LPDR, to support the clinical interpretation of the 161 unpublished variants, including family history data and clinical measurements. They assessed the pathogenicity of the variants using ACMG and the Association for Molecular Pathology (AMP) guidelines [18].

The analysis revealed 48 (29.8%) variants classified as “Pathogenic” and another 91 (56.5%) classified as “Likely Pathogenic”. Eleven (11) variants were discovered to be incorrectly reported and were not further analyzed. Another 11 had inadequate clinical information to assign a classification and were considered Variants of Uncertain Significance (VUS).

These findings may aid in confirmatory clinical testing after a positive newborn screen and demonstrate the utility of the data sets contained in the LPDR to advance genetic research in the NBS setting.

##### BabySeq Use Case

The BabySeq project was part of the (later named) Newborn Sequencing In Genomic Medicine and Public Health (NSIGHT) consortium to explore how genomic information could be used to inform the identification of diseases in newborns. BabySeq was a randomized control trial of exome sequencing in healthy and sick infants with the return of results for childhood-onset diseases to parents and the newborn’s healthcare provider [19].

Early on, the consortium initiated a CDE workgroup to identify and standardize the collection of common phenotypic data elements across the four sites. It was decided to store the common phenotypic data elements and the genomic data in one place—in the NBSTRN LPDR—to facilitate genetic research across the four sites and across the studies contributing to the LPDR. In addition, it facilitated the NBSTRN, generating periodic reports on what data had been collected by the NICHD/NHGRI and allowed the NBSTRN to contribute our data to other efforts, including dbGaP. All of the BabySeq phenotype and genotype data are now in the NBSTRN’s LPDR for future research.

BabySeq2 has now been launched and includes three enrollment sites (Boston Children’s Hospital, the Icahn School of Medicine at Mount Sinai in New York, and the University of Alabama at Birmingham). The goal is to enroll 500 babies from clinics that serve under-resourced African American and Hispanic communities in a randomized control trial of genome sequencing with the return of results to parents and the babies’ healthcare providers. The genomic sequencing data and CDEs will be added to the LPDR, building the NBSTRN resource even further.

##### BeginNGS Use Case

The recently launched BeginNGS™ (Begin Newborn Genome Screening), led by Rady Children’s Institute for Genomic Medicine (RCIGM), is a healthcare delivery system designed to screen newborns for genetic diseases and connect their doctors with effective interventions [20]. Through a public–private consortium of leading organizations and advocacy groups in pediatrics, genetics, biopharma, biotech, and information technology, BeginNGS is developing and piloting a system for parentally consented, artificial intelligence-based, and genome sequence-based screening and virtual acute management guidance for at least 500 severe childhood genetic diseases that have effective therapies. BeginNGS utilizes multiple NBSTRN resources including the LPDR, NBS-CR, ELSI Advantage, and NBS Genomics Resource (NBS-GR) [21]. As the pilot proceeds, the genomic sequencing data gathered will be deposited into the LPDR, further building the NBSTRN resource.

Other data available from the LPDR to facilitate future clinical research include:Common phenotypic data elements and genomic data from the NSIGHT1 study of rapid whole-genome sequencing for diagnosis in critically ill infants [22].Common phenotypic data elements and genomic data from the NSIGHT2 study of the analytic and diagnostic performance of rapid whole genome and exome sequencing in ill infants [23].

### 3.2. NBS Conditions Resource

The NBSTRN created the NBS-CR to provide a centralized source for information and statistics for NBS conditions. NBS-CR currently includes 35 RUSP core conditions, 26 RUSP secondary conditions, and 73 conditions identified by the NBSTRN as candidates for pilot studies [24]. Some of the 73 candidates have been previously nominated to the RUSP but are not yet recommended for listing, and some are included in state-level NBS panels. Because state panels are continuously evolving, individual state NBS programs are the definitive source of the NBS status of a particular condition at a specific point in time. The NBS-CR is designed to be an interactive resource for researchers, clinicians, parents, and families to learn more about these disorders.

#### 3.2.1. Stakeholder Resources

The NBS-CR provides resources for the array of NBS stakeholders.

The Genome Data Viewer visualizes the location of the affected gene within the genome and on the specific chromosome. A link takes the reader to deeper genomic information at the National Library of Medicine [25].An overview of the condition’s genetic cause, onset, symptoms, and pathophysiology.Resources for researchers and clinicians with condition-specific links to sources of general, clinical, research, and further reading information.Resources for the general public, including a link to the genetic condition’s information at MedlinePlus [26].

The database is updated as new information and data become available. The NBS-CR has been described by some as a treasure trove of information for researchers, clinicians, state public health programs, families, advocacy groups, and more. Another important resource developed by the NBSTRN is the monthly webinar to facilitate information sharing between state newborn screening programs, researchers, clinicians, advocates, and federal partners about conditions recently added to the RUSP, conditions currently part of pilots, or new candidate conditions that could be considered for screening as described in the NBS-CR. These webinars are recorded and are publicly available on the NBSTRN YouTube channel (https://www.youtube.com/@nbstrn (accessed on 4 February 2023)).

#### 3.2.2. NBS-CR Use Cases

##### Early Check Use Case

Early Check is a voluntary expanded NBS study available to babies born in North Carolina. The project is led by researchers at RTI International in partnership with the University of North Carolina at Chapel Hill and the North Carolina State Laboratory of Public Health [27]. Since 2018, over 26,000 newborns have been screened for a targeted panel of conditions. In the next phase of Early Check, parents will be offered genome sequencing-based screening for two panels of rare monogenic conditions and the risk of developing type 1 diabetes. Monogenic panel 1 will include conditions with highly effective interventions/therapies and panel 2 will include conditions with therapies that are less effective or therapies in clinical trials.

Early Check strives to include conditions in the panels that are not currently identified by state NBS but are being championed by patient advocates, parents, and clinicians for addition. Having access to centralized information about pilot candidates within the NBS-CR significantly simplifies the identification of these conditions. Within the NBS-CR users can filter conditions by nomination status (RUSP-Core, RUSP-Secondary, and/or candidate) and download information about the selected conditions. The Early Check team reviews each candidate condition to determine if it meets inclusion criteria (actionable before age 2 years) and could feasibly be diagnosed by genome sequencing. From the NBS-CR candidate list, 18 conditions (22 genes) were added to panel 1 and 15 conditions (16 genes) were added to panel 2. Candidate conditions that were not included were either not genetic (e.g., congenital cytomegalovirus infection), not detectable by genome sequencing (e.g., Prader-Willi syndrome), not validated for reporting by our laboratory (e.g., Fragile X syndrome), or did not meet our actionable by age 2 criteria (e.g., familial hypercholesterolemia). Ultimately, the availability of the NBS-CR ensured that Early Check was able to include conditions of high interest to many in the NBS community.

##### Family Narratives Use Case

Bush et al. [28] conducted retrospective research using a relatively large qualitative interview analysis approach to highlight the voices of parents already caring for children identified with NBS-related conditions as characterized in the NBS-CR. The narratives of these caregivers richly describe the lived experiences of their families as they share their journey through the “diagnostic odyssey continuum”, a term informally used by Bush since the 1980s in the context of how families navigate the uncertain course of how their child is “labeled”—from presumptive diagnosis, a confirmatory diagnostic “label”, and uncertainties in prognosis and management. This research, employing a subset of data from Koehly’s “Inherited diseases, caregiving, and social networks” NIH intramural protocol, focuses on the diagnostic odyssey continuum of NBS-related conditions using the NBS-CR in light of current research by many NBSTRN investigators examining NBS expansion with genomic technologies.

The insights offered by the families who care for children with conditions included in the NBS-CR illustrate that the course from screen to diagnostic labels is often an evolving process, particularly when accompanied by prognostic uncertainty. Such perspectives emphasize the importance of managing expectations of phenotypes, medical/educational/social needs, outcomes, quality of life, and caregiving responsibilities. The issues raised by parent caregivers provide insight on the path to maximizing beneficial interests for the child and minimizing harm, thus easing the burden for families of children with NBS-identified conditions. This information aims to contribute to the development of ethically nuanced policy as the expansion of genomics accelerates in population-based newborn public health programs.

### 3.3. NBS Virtual Repository of States, Subjects, and Samples

The NBSTRN created the NBS-VR to provide state-specific NBS program details in response to the fact that there are 53 non-federal NBS programs in the U.S.: 50 states, 2 territories, and the District of Columbia. Each NBS program establishes its own policies and procedures, including [29]:Which conditions to screen;If and how to store residual dried blood spots (DBS);Whether to obtain consent from parents for the use of DBS;Whether to conduct a long-term follow-up of diagnosed cases.

The NBS-VR helps to facilitate collaboration among federal and state NBS programs.

#### NBS-VR Tools

The NBS-VR has an interactive map of the U.S. from which users can select several different options and data types:National or individual statesConditionScreening implementationEstimate of cases per yearAnnual birthsNBS program information

A companion interactive table provides similar data options and can be viewed or downloaded in summary form or detailed form.

Both the map and table allow the exploration of different aspects of national and state NBS programs, including:Screening adoption—Explores the makeup of state screening panels with filters by RUSP and/or candidate status, ACHDNC category, or individual condition.Expected cases—Explores the number of cases based on published disease incidence with filters by nomination status, ACHDNC category, or individual condition.Demographic information—Explores the annual number of births by the NIH racial and ethnicity categories.National NBS program information—Explores the Regional Genetic Networks, Dried Blood Spot storage, retention time and consent, and longitudinal follow-up.State NBS program information—Explores information including a link to the NBS program website, NBS program contacts, and second screen policy.

Thus, the NBS-VR tool makes information on screening practices available to and manageable by all NBS stakeholders.

### 3.4. Ethical, Legal, and Social Issues (ELSI) Advantage

The NBSTRN created the ELSI Advantage tool to help the NBS research community as they consider ELSI that may arise in the planning and implementation of research studies [30]. The tools included in ELSI are:ELSI 101 summarizes topics related to ELSI in NBS research. The content is developed and maintained by the NBSTRN Bioethics and Legal Workgroup.Ask ELSA! is an interactive tool where users can ask questions about ELSI and NBS research. The responses are drawn from a database of ELSI topics in NBS.The research repository is a searchable, curated resource that summarizes and links to key publications that address ELSI in NBS.The policy map is an interactive tool that provides information on ELSI topics and state NBS program policies and procedures.Users can schedule a consultation with the NBSTRN Expert Workgroup to discuss ELSI and NBS research topics.The video library provides access to NBSTRN’s ELSI-related video content, such as webinars, network meetings, research summits, and more.

ELSI Advantage content is also useful for many other NBS stakeholders that encounter ELSI considerations such as healthcare professionals, families, advocacy groups, etc.

The ELSI Advantage provides a valuable resource for anticipating, considering, educating, and addressing some of the complexities and tensions that arise with ELSI in public health, research, and clinical domains. This extends from conceptualizing research studies in the early phase, through the analysis, and to teaching multi-disciplinary healthcare providers.

The International Journal of Neonatal Screening’s recent Special Issue, “Ethical and Psychosocial Aspects of Genomics in the Neonatal Period” [31], contains 15 articles that address a multitude of NBS ethical considerations and garnered the attention of many in the NBS community. ELSI Advantage considerations were discussed throughout the issue.

Other reports on NBSTRN’s ELSI work include:Including ELSI research questions in newborn screening pilot studies [32];Duchenne Muscular Dystrophy Newborn Screening, a Case Study for Examining Ethical and Legal Issues for Pilots for Emerging Disorders: Considerations and Recommendations [33].

## 4. Discussion

The NBSTRN provides an array of tools that aggregate research data, CDEs, condition-specific information, state NBS program details, and ELSI considerations. The tools are accessible on the NBSTRN website, and each has a user interface that makes the tool easy to navigate and use. The diverse NBS stakeholders can use the tools to obtain data and information to support their research, clinical practice, children, families, and communities.

The LPDR acts as a nationwide platform where pilot and longitudinal data and CDEs are aggregated, including that from state NBS pilots, and made available to the NBS research community. NBS researchers and state NBS programs use the LPDR to access data and other information to assist in study design, data evaluation, and NBS program development. The LPDR is also an important tool in helping to support and accelerate the RUSP nomination and evaluation process.

Current LPDR data are but a fraction of what has been generated by all NBS researchers. Every set of data is, however, crucial to the growth of robust data sets, and knowing what additional data are needed can help the NBS research community design further studies that will help fill in the gaps. We encourage researchers, clinicians, and other stakeholders to contact the NBSTRN to discuss including their data in the LPDR. This type of cooperative mindset and effort will enable faster, more targeted development of screening protocols and treatments that can provide babies and their families with better outcomes.

In addition, the NBSTRN is in the process of developing the NBS-GR tool [21]. NBS-GR will serve as a centralized repository of the conditions that are currently being studied in research efforts exploring the use of novel technologies, including genome sequencing, in NBS. The tool’s purpose is to assist the NBSTRN community of researchers, clinicians, parents, and families to track the conditions and genes that are being investigated and provide information from various projects and programs as they develop.

## Data Availability

No new data were created, and thus, this statement is not applicable.

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
