# Peer review of "NBSTRN Tools to Advance Newborn Screening Research and Support Newborn Screening Stakeholders"

_2409-515X, 2023, doi:10.3390/ijns9040063_

Round 1

Reviewer 1 Report

Comments and Suggestions for Authors

The work proposed by Chan et al raise the important question of the "tools" developed to help the stakeholders toward extension of NBS programs.

The paper is very interesting and the only minor point is that all the paper describe the US context and the question of NBS extension is often very specific of each country. May be the authors should add a paragraph to explain this point in the introduction and to discuss their results and their "adaptation" to others countries

Author Response

We appreciate Reviewer 1 review and suggestions of our manuscript.

To address the comment, "May the authors should add a paragraph to explain this point in the introduction and to discuss their results and their "adaptation" to others countries", we have included the following changes in the revised manuscript and included additional references below: 

Line 86 - 114

Further, most NBS conditions represent a continuum of disease severity with variability in clinical presentation and outcome.  For example, in phenylketonuria (PKU), plasma phenylalanine (Phe) levels and tolerance to ingested Phe vary widely among affected individuals, with the milder end of the spectrum typically not requiring treatment [5]; while, other conditions such as the rare Pompe disease are variable rates of disease progression and different ages of onset.. 

Outside the US, most NBS programs in Europe have modernized the screening process with new technology or algorithms with new information (i.e. modifying cut-off rate based on different platforms) and expanded their panel of conditions screened based on the screening criteria in different countries [6] . However, the decision-making oversight across the globe varies in their adoption, implementation, and resources to support short and long-term management of care. To facilitate the advancement and sharing of NBS research, the International Society of Neonatal Screening is in the process of establishing a database which is available for public consultation [6]. As the cost of genome sequencing has decreased drastically over the last decade and the speed of using genomic data to inform diagnosis have increased, many countries in Europe, Australia and China are considering the application of genomic sequencing platform to complement newborn screening [7–9].  

This paper describes the data tools and resources developed and implemented by NBSTRN to facilitate NBS research and support NBS stakeholders locally and internationally. NBSTRN’s web-based tools are accessible worldwide to share knowledge and advance the development and expansion of NBS programs locally and internationally. We provide use cases for each NBSTRN tool such as the NBS-Condition Resources (NBS-CR) for information about screenable disorders, dissemination of results from pilot programs and to identify candidate conditions for newborn screening across the globe. For example, international countries could use the NBS-CR to identify possible candidates for newborn screening. We also discuss new tools that are currently not available and that may be required as NBS expands to include a larger number of genetic disorders that are screened in increasingly diverse populations.

We also cited the additional references in the revised manuscripts: 

Adams, A.D.; Fiesco-Roa, M.Ó.; Wong, L.; Jenkins, G.P.; Malinowski, J.; Demarest, O.M.; Rothberg, P.G.; Hobert, J.A.; ACMG Therapeutics Committee. Electronic address: [email protected] Phenylalanine Hydroxylase Deficiency Treatment and Management: A Systematic Evidence Review of the American College of Medical Genetics and Genomics (ACMG). Genet Med 2023, 25, 100358, doi:10.1016/j.gim.2022.12.005.

 Loeber, J.G.; Platis, D.; Zetterström, R.H.; Almashanu, S.; Boemer, F.; Bonham, J.R.; Borde, P.; Brincat, I.; Cheillan, D.; Dekkers, E.; et al. Neonatal Screening in Europe Revisited: An ISNS Perspective on the Current State and Developments Since 2010. IJNS 2021, 7, 15, doi:10.3390/ijns7010015.

Wiley, V.; Webster, D.; Loeber, G. Screening Pathways through China, the Asia Pacific Region, the World. Int J Neonatal Screen 2019, 5, 26, doi:10.3390/ijns5030026.

Stark, Z.; Scott, R.H. Genomic Newborn Screening for Rare Diseases. Nature Reviews Genetics 2023, 1–12.

Jiang, S.; Wang, H.; Gu, Y. Genome Sequencing for Newborn Screening—An Effective Approach for Tackling Rare Diseases. JAMA Netw Open 2023, 6, e2331141, doi:10.1001/jamanetworkopen.2023.31141.

Reviewer 2 Report

Comments and Suggestions for Authors

This report summarises a really important initiative to support newborn screening programs in the USA in evaluation of disorders on, or potentially on, newborn screening panels.

The authors might like to mention also the regular webinars sharing information about disorders and pilot programs.

It would also be helpful if the authors could comment on whether there is any potential for international use of the databases / international contributions to data and if so how they might be accessed.

Two teeny weeny quibbles -

P2 l51 note programs count disorders in different ways and these numbers are likely not comparable.

P2 l72 all nbs conditions are variable, issue arises when there is most variability and some is significant but late onset (eg pku is variable, but the mildest won’t present with issues caused by mild hyperphe in late childhood or adulthood so the variability isn’t an issue as it is in eg Pompe.

Author Response

We appreciate Reviewer 2 review and suggestions for our manuscript. To address the comment " The authors might like to mention also the regular webinars sharing information about disorders and pilot programs", we have included the following paragraph:

Line 315 - 321

Another important resource developed by NBSTRN is the monthly webinar, to facilitate information sharing between state newborn screening programs, researchers, clinicians, advocates, and federal partners about conditions recently added to the RUSP, conditions currently part of pilots or new candidate conditions that could be considered for screening as described in the NBS-CR.  These webinars are recorded and are publicly available on the NBSTRN YouTube channel (https://www.youtube.com/@nbstrn).

To address the comment, "It would also be helpful if the authors could comment on whether there is any potential for international use of the databases / international contributions to data and if so how they might be accessed,". we have included the following changes: 

Line 95-99

However, the decision-making oversight across the globe varies in their adoption, implementation, and resources to support short and long-term management of care. To facilitate the advancement and sharing of NBS research, the International Society of Neonatal Screening is in the process of establishing a database which is available for public consultation [6].   

To address the comment, " P2 l51 note programs count disorders in different ways and these numbers are likely not comparable," we included the following sentence in the revised manuscript: 

"(line 54-55). It is important to note that state programs count disorders in different ways and these numbers are likely not comparable."

To address the comment " P2 l72 all nbs conditions are variable, issue arises when there is most variability and some is significant but late onset (eg pku is variable, but the mildest won’t present with issues caused by mild hyperphe in late childhood or adulthood so the variability isn’t an issue as it is in eg Pompe," we included the following sentence in the manuscript, " (line 86-91), 

Further, most NBS conditions represent a continuum of disease severity with variability in clinical presentation and outcome.  For example, in phenylketonuria (PKU), plasma phenylalanine (Phe) levels and tolerance to ingested Phe vary widely among affected individuals, with the milder end of the spectrum typically not requiring treatment [5]; while, other conditions such as the rare Pompe disease are variable rates of disease progression and different ages of onset."
